# Exploring Secondary School Athletic Trainers’ Perspectives in Managing Mental Health Situations

**DOI:** 10.3390/ijerph21050577

**Published:** 2024-04-30

**Authors:** Suhyun Jang, Matthew J. Drescher, Tara A. Armstrong, Elizabeth R. Neil, Lindsey E. Eberman

**Affiliations:** 1Department of Applied Medicine and Rehabilitation, Indiana State University, Terre Haute, IN 47809, USA; lindsey.eberman@indstate.edu; 2Health, Nutrition, and Exercise Sciences, North Dakota State University, Fargo, ND 58102, USA; matthew.drescher@ndsu.edu; 3Athletics-Sports Medicine, Drake University, Des Moines, IA 50311, USA; tara.armstrong@drake.edu; 4College of Public Health, Temple University, Philadelphia, PA 19122, USA; beth.neil@temple.edu

**Keywords:** secondary school student–athletes, preparedness, confidence level, mental health management policies and guidelines, employment structures

## Abstract

Background: Most people believe that student–athletes experience fewer difficulties related to mental health than non-sport participants. However, several studies have shown high depression levels or emotional difficulties in adolescent athletes. Most secondary school students have access to athletic trainers in their schools. Secondary school athletic trainers (SSATs) are medical professionals who can provide health care for student–athletes, and they could be appropriate supporters in providing mental health management in secondary schools. However, there are no studies that have addressed their preparedness and confidence level to manage potential risk factors and mental health disorders. This study aims to ascertain preparedness and confidence levels from SSATs’ perspectives in handling mental health disorders using a survey based on the mental health management guidelines and consensus statement. Methods: This is a cross-sectional study design. The research team created an online survey questionnaire based on the National Athletic Trainers’ Association (NATA) Mental Health Guidelines for Secondary School and Interassociation Recommendation: A Consensus Statement. Utilizing the NATA Research Survey Service, the online survey was emailed to SSATs who self-categorized as secondary school athletic trainers in the NATA membership system. (*n* = 171, 65% completion rate). SPSS was used to analyze the survey data. Result: This study found that 29.2% of SSATs reported they have policies or guidelines regarding mental health disorders management for minors. The most frequent policy was mandatory reporting in cases in which an individual is being abused or neglected (80.5%). The highest confidence area was aligned with the most frequent policy. The least frequent policy was considering unique stressors and triggers with student–athletes to recognize the potential mechanisms that may cause a mental illness or exacerbate an existing mental illness (58.5%). Even though the least frequent policy was not aligned with the least confidence area, it was the second lowest confidence level. This study identified significant differences between two different employment structures: school-based employment and hospital-based employment structures. Policies were more common in school-based employment than in hospital-based employment structures. Conclusion: This study demonstrated the gap between organizational and individual preparation. The majority of SSATs respondents expressed moderate to high confidence in their ability to manage mental health disorders, despite the lack of mental health policies or procedures in their secondary schools. This study recommends that SSATs create guidelines or procedures in the areas where they are least confident and prepared to offer mental health management for minor student–athletes. They can use the interassociation recommendations and the NATA guidelines for mental health care to close the gap.

## 1. Introduction

One study provided the national level of the prevalence of treatable mental health disorders in US adolescents aged 0 to 17 years using data from the 2016 National Survey of Children’s Health. Based on this study, 7.7 million children have at least 1 mental health disorder out of an estimated 46.6 million children. It revealed that half of the estimated 7.7 million US children with treatable health disorders did not receive needed treatment from a mental health professional [1]. Unfortunately, the prevalence of emotional and behavioral disorders in children is as common as that of physical conditions, such as asthma and diabetes [2,3,4]. The definition of a mental health disorder includes a broad meaning. The American Psychiatric Association Diagnostic and Statistical Manual of Mental Disorders, fifth edition, states that “A mental disorder is a syndrome characterized by clinically significant disturbance in an individual’s cognition, emotional regulation, or behavior that reflects a dysfunction in the psychological, biological, or development process underlying mental functioning” [5]. In this study, mental health disorders refer to mental health conditions, including anxiety, behavior, depression, or eating disorders, that could be observed in secondary schools. In addition, possible risk factors or triggers that could affect psychological health concerns are included. Based on previous studies, people might believe that student–athletes have lower perceived stress, lower depression levels, and higher self-rated mental health than non-sport participants [6,7]. However, high depression levels, negative psychological reactions, or emotional difficulties have been observed in student–athletes [4,5,8]. These findings contradict previous studies that student–athletes have better coping strategies for mental health difficulties than non-sport individuals [6,7,9]. Among adolescents in the United States, the most identified mental health condition was anxiety disorders (31.9%), followed by behavior disorders (19.1%), mood disorders (14.3%), and substance use disorders (11.4%) [2].

The American Medical Association has indicated that ATs are the most appropriate medical professionals to provide health care for secondary school student–athletes [10]. ATs are allied healthcare professionals qualified to provide various healthcare services, such as injury prevention, examination, clinical diagnosis, and emergency case management. ATs contribute to minimizing injury risks and returning student–athletes to play as soon as possible [11]. Unlike other healthcare professionals, ATs spend a great deal of time with patients, from when they are injured to when they return to play. Therefore, building rapport and trust is one of the crucial abilities for ATs. A cross-sectional study assessed ATs’ abilities to recognize psychological signs, choose suitable interventions, and make referral decisions to appropriate providers. The study demonstrated that ATs showed high accuracy in identifying symptoms and making referral decisions [12]. Over half (66%, *n* = 13,473/20,272) of secondary school students have access to athletic training services [13]. Secondary school athletic trainers (SSATs) can be a critical tool for addressing the growing mental health disorders among secondary school student–athletes’ health. However, recent studies have shown that SSATs spend most of their time at practices, game coverage, emergency planning, injury prevention, concussion reporting, rehabilitation, and injury treatment, which limits the available time to spend in the clinic addressing mental health disorders [11,14,15]. Patients who received social support from SSATs reported that such interactions had a positive impact on their physical and psychological recovery [11,16]. ATs should be able to provide whole-person health care, including the recognition and management of mental health disorders, but barriers, both real and perceived, might limit their ability to provide these services. The development of policies and procedures related to the management of mental health disorders may provide SSATs with support to manage these cases.

The current study used previous research that assessed the mental health policies and procedures implemented and collegiate ATs’ perceived confidence in managing mental health disorders for all three divisions of NCAA athletes [17]. Although there are a few studies that showed the confidence levels of ATs in managing mental health disorders [18,19], no articles showed the existence of policies and guidelines that demonstrated the provision of mental health management services in secondary school settings, even though secondary school student–athletes have shown difficulties with psychological issues [4,5,8]. There are guidelines and consensus statements that provide for mental health care procedures, recognition, and referral plans for SSATs. The National Athletic Trainers’ Association (NATA) published an emergency action plan guideline for mental health emergencies in secondary schools in 2016, and an interassociation consensus statement was released by the NATA in 2015 [4,5]. However, there are no studies that detail whether SSATs possess policies, procedures, or guidelines for mental health management in their schools, and there are no studies that identify SSATs’ preparedness, knowledge, and confidence levels in mental health disorders management. This study aimed to investigate preparedness and confidence levels in managing mental health disorders from the perspective of SSATs utilizing questionnaires based on the current mental health management guidelines and consensus statements.

## 2. Materials and Methods

We used a cross-sectional design to study SSATs’ preparedness and confidence levels associated with mental health disorders management. This project received an exempt determination from the Institutional Review Board at Indiana State University, and all participants provided informed consent before engaging in this study.

### 2.1. Participants

The inclusion criteria of this study were part-time and full-time certified ATs who self-categorized as SSATs in the NATA membership system. The NATA Research Survey Service was used to recruit SSATs through emails. It distributed the online survey via Qualtrics^®^ to the subject population. The survey was distributed for four weeks in two recruitment cycles in April/May and October 2023. Reminder emails were sent every week for each of the four weeks. We sent a total of 3083 emails, to which 263 participants responded, and 171 participants completed the survey. It results in an 8.5% access rate and a 65% completion rate. Participants were an average of 41 ± 12 years old with 17 ± 11 years of experience (Table 1).

### 2.2. Instrumentation

The online survey tool included questionnaires related to the preparedness and confidence level of SSATs in mental health management and was created based on the NATA Mental Health Guidelines for Secondary School [4] and Interassociation Recommendation: A Consensus Statement [5]. This 47-item questionnaire (Table 2) included the following: school demographics (11 items), guidelines and recommendations in mental health management (16 items), confidence in mental health management (15 items), and previous mental health management experiences (5 items). The school demographics included employment structures (school-based and hospital-based) and employment characteristics (full-time split with the health care unit, full-time split with the academic unit, part-time with school). In the demographics, school district employment and school district employment with teaching responsibilities employee categories were included in the school-based employment structures. Hospital-based employment structures consisted of hospital or clinic employees. The survey asked for background information on the mental health policies or guidelines used at the participant’s clinical practice site, including when those policies or guidelines are reviewed and drafted, which groups/organizations are involved in creating them, as well as demographic questions. The online survey provided explanations of the guidelines or recommendations and asked if their clinical sites have policies or guidelines related to the specific recommendations. Finally, participants were asked to rate their confidence levels in acting on the related policies or guidelines in their clinical practices. The confidence question items were placed on a sliding scale of 0–100 (0 = No confidence, 100 = Full confidence). Previous studies supported the notion that the scale has several advantages. The scale has shown excellent validity and reliability when measuring confidence levels [20]. In addition, participants were asked to provide the number of mental health or potential violence cases managed in the past 12 months.

Three content experts used a Content Validity Index (CVI) to evaluate the instrument. The scale-level CVI (S-CVI) of 0.80 or higher indicates that the instrument items are to be quite or highly relevant [21]. After the research team reviewed and revised the survey questionnaire based on the content experts’ rates, S-CVI was deemed to have high content validity (S-CVI/Ave. = 0.96).

### 2.3. Statistical Analysis

The survey results were analyzed using the SPSS (IBM Corp. Released 2021. IBM SPSS Statistics for Windows, Version 28.0. Armonk, NY, USA) statistical software. The research team conducted a statistical analysis using descriptive analysis for demographic characteristics and to summarize the presence of policies, procedures, and guidelines in mental health management. Partial data were used for analysis. A Chi-square analysis was used to identify the differences in preparedness between the levels of educational degrees, employment structures, and characteristics of employment associated with mental health management policies, procedures, and guidelines. A Mann–Whitney U test was used to identify differences in the existence of policies between employment structures and employment characteristics.

## 3. Results

A total of 171 SSATs completed the survey. The respondents’ mean age was 40.7 *±* 12.3, with 62.6% female participants (107/171) and 36.8% male participants (63/171). Additional demographic data are provided in Table 1. Based on the survey, 29.2% of SSATs reported they have policies or guidelines managing mental health disorders for minors. The existence of policies in alignment with the recommendations varied between 14.7% (planning to deal with the psychological concerns considered legal implications) and 80.5% (mandatory reporting relative to abuse or neglect) (Table 2). The three most frequently present policies included the following: (1) having a policy related to mandatory reporting in cases in which an individual is being abused or neglected in any way (80.5%); (2) having a policy related to mandatory reporting in cases in which an individual poses a risk to themself or others in any way (71.9%); (3) having a policy that includes considerations for the county, state, and federal laws and regulations regarding mandatory reporting (67.3%). Participants reported the highest confidence levels related to these three policies (Table 2). The three least frequently present policies included the following: (1) having a policy for considering unique stressors and triggers with student–athletes to recognize potential mechanisms or events that may create a mental illness or exacerbate an existing mental illness (58.5%); (2) having a policy for reviewing mental illness in student–athletes to recognize potential psychological concerns (54.7%); (3) having a policy for stakeholders to develop monitoring strategies to recognize concerning behavior changes that could be related to mental health special considerations such as changes in eating and sleeping habits and unexplained weight loss or weight gain (51.5%). Participants reported moderate confidence in these areas (Table 2).

The level of confidence also varied across the recommendations between 46.6% (considering legal implications in developing a plan) and 81.7% (mandatory reporting relative to abuse or neglect) (Table 2). The three policies that participants were most confident in implementing were the following: (1) mandatory reporting in cases in which an individual is being abused or neglected in any way (81.7%); (2) mandatory reporting in cases in which an individual poses a risk to themself or others in any way (80.5%); (3) executing duty as a mandatory reporter given the county, state, and federal laws and regulations (77.5%). The policies that participants were least confident in implementing were the following: (1) considering legal implications in developing a plan to manage the psychological concerns of student–athletes (46.6%); (2) reviewing mental illness in student–athletes to recognize potential psychological concerns (54.1%); (3) considering unique stressors and triggers with student–athletes to recognize potential mechanisms or events that may create a mental illness or exacerbate an existing mental illness (57.1%). (Table 2).

We did not identify any significant difference between groups for the highest degree earned and characteristics of employment on the presence of or levels of confidence for the recommended policies, procedures, or guidelines. We identified significant differences between employment structures (Table 2): referral (*p* = 0.033), recognizing specific indicators of mental health special considerations (*p* = 0.009), monitoring strategies (*p* = 0.001), stakeholder education (*p* = 0.040), resource availability (*p* = 0.006), mandatory reporting relative to the county, state, and federal regulations (*p* < 0.001), mandatory reporting for potential self-harm or harm of others (*p* = 0.002), mandatory reporting for potential abuse and neglect (*p* = 0.003), activating the chain of command for campus crisis (*p* < 0.001), the training of stakeholders in recognizing stressors that may lead to violence (*p* = 0.015), referral for potential violence to mental healthcare providers (*p* = 0.005), and an MH emergency action plan (*p* = 0.021). School-based employment structures were more likely to have policies compared to hospital-based employment structures (Table 2).

Similarly, we also identified significant differences between employment structures on the level of confidence (Table 2): collaborating with other stakeholders (*p* = 0.036), mandatory reporting relative to the county, state, and federal regulations (*p* = 0.049), mandatory reporting for abuse and neglect (*p* = 0.013), activating the chain of command for campus crisis (*p* = 0.009), and recognition and referral for mental health emergencies (*p* = 0.020). In all recommendation areas, school-based athletic trainers rated higher levels of confidence than hospital-based athletic trainers.

## 4. Discussion

Mental health disorder cases in student–athletes have been well observed in secondary schools. Healthcare professionals and stakeholders working in these settings should consider possible psychological responses from this population [5,8]. As an allied healthcare professional, SSATs could encounter student–athletes with mental health disorders, and they should have the ability to recognize behavior changes, unique stressors, and legal considerations so that they can assist student–athletes who are suspected of having mental health disorders. In addition, SSATs should be able to educate student–athletes, coaches, and stakeholders about the possible risk factors or triggers that could affect psychological health concerns by collaborating with school personnel [4,5]. These are crucial strategies to prevent possible emergency circumstances. For this reason, critical incident management is delineated as a practice standard of all athletic trainers, including SSATs, and all should have in-depth mental healthcare-related policies or guidelines in their schools [22]. No articles, however, have identified whether SSATs are prepared to manage mental health disorders for minors. This study sought to address the preparedness and confidence level of SSATs in managing psychological instabilities. This study showed a disconnection between organizational and personal preparedness, where SSATs showed moderate to high confidence levels in managing mental health, despite most of them reporting they do not have or do not know whether they possess policies or guidelines for mental health management. This study also identified significant differences in employment structures for the presence of recommended policies. Specifically, school-based employment structures were more likely to have policies than hospital-based ones.

### 4.1. The Overall Presence of Policies

Several studies stated the importance of policies and guidelines for mental health disorders to provide appropriate care for collegiate student–athletes or over 18-year-old athletes [16,23,24]. Also, there is a study that assesses the mental health policies and procedures implemented in managing mental health disorders for the National Collegiate Athletic Association (NCAA) institutions [17]. This is the first study to review the preparedness of mental health management for secondary schools by assessing current implemented policies or guidelines. This research states that 29.2% of SSATs reported they have policies or guidelines managing mental health disorders for minors. The most common policies were related to mandatory reporting in terms of mental health disorders. It showed that most of the mental health policies of SSATs have included mandatory reporting in psychological concern management. The least common policies were associated with prevention strategies, including recognizing potential concerns that could cause mental health disorders, educating stakeholders about monitoring, and recognizing behavior changes. Mental health is likely to be affected by environmental factors, such as gender issues, bullying, or transition from sports [24]. Since those factors may not be apparent in student–athletes’ behaviors, collaborating with healthcare professionals and educating stakeholders are important strategies to recognize or monitor risk factors and prevent mental health disorders. Therefore, having related policies or guidelines is encouraged in schools even though those are not mandatory legal requirements according to the county, state, and federal regulations.

### 4.2. The Overall Confidence

This is the first study to examine SSATs’ confidence in the implementation of policies related to mental health disorders. Previous research has examined the confidence of ATs in managing mental health disorders, however, the findings have not been consistent [18,19]. ATs indicated moderate to extreme confidence in managing patients with psychological conditions in one study [18], but others stated low confidence in managing patients with eating disorders in another study [19]. However, neither of these studies examined SSATs specifically. Because of the impact of mental health disorders on the health of adolescents, it is imperative that SSATs feel confident in implementing policies to manage such conditions. This study identified that SSATs were very confident in acting as mandatory reporters for patients with mental health disorders. This may be due to the legal requirement of the mandatory reporting of behavioral health concerns in many states. The legal requirement for reporting and the fear of liability may negate the ethical dilemma of reporting these situations. Further, individuals who interact with adolescents receive frequent training on their roles and responsibilities as mandatory reporters, which may give them higher confidence in intervening for at-risk patients. SSATs, however, indicated lower confidence in identifying risk factors for mental disorders and in training others to recognize these factors and special considerations. The identification of these risk factors, without proper training, can be nebulous and subjective, and SSATs may not be provided with adequate training to feel confident in identifying risk factors [12,25]. The literature has shown that even brief training can improve the confidence and skills of nurses to manage patients with suicidal ideation [26]. The implementation of training for SSATs to identify risk factors for mental health disorders, as well as training on how to educate stakeholders, may help to improve confidence in identifying these risk factors as well.

### 4.3. Employment Structure

As we discussed above, we encouraged having policies and guidelines for mental health disorder management. We also suggested having adequate training to improve on the low confidence level sections. In order to improve those aspects, we also should consider organizational and management structures. The quality of health care can be affected by organizational structures and attributions [27,28]. It could include different elements, such as executive management, culture, or organizational design. A study demonstrated that a hospital with a centralized design, in which each functional department has a manager who is required to report to higher management levels, has a beneficial efficiency in terms of economy. However, this design could inhibit the ability to improve innovation and creative quality improvement processes at the level of the practice line [29]. Depending on organizational management structures, it could maximize the ability of individuals, which results in the quality of patient care in the health profession [27,28]. This study looked at differences in the existence of policies in employment structures, which were divided into school-based and hospital-based employment structures. It showed there were significant differences between employment structures even though there were no significant differences between employment characteristics. This study showed that policies were found to be more prevalent in school-based employment structures than in hospital-based employment structures. Because adolescent mental health is a substantial component of the organization’s service, schools may be more likely to have policies than hospital systems, where AT service is a small part of the entire hospital organization. This study encourages hospital-based SSATs to possess policies or guidelines to prevent inherent risks that are relevant to psychological disorders.

### 4.4. Limitations and Future Research

One limitation of this study is the small sample size. We emailed 3083 SSATs who are currently NATA members, and only 171 of them participated. This small sample size may not fully represent the population. In addition, this survey was only distributed to NATA members who self-categorized as SSATs. However, there are a large number of SSATs who are not NATA members. Another limitation is the accuracy of the reported presence of policies and guidelines. This study did not directly verify the existence of mental health-related policies and guidelines for all participants and, therefore, relied solely on their responses to survey questions. Future studies may want to consider increasing the sample size to be more representative of the population. In order to increase the study’s accuracy, similar studies could be conducted in specific counties or school districts by asking SSATs to upload their policies and procedures. In addition, future studies could extend this study using demographic variability, such as years of experience, to determine differences in the preparedness and confidence level between SSATs who are new to this setting and those who are more experienced.

## 5. Conclusions

This study showed the disconnect between organizational and personal preparedness. Although most SSATs reported the lack of mental health policies or guidelines’ existence in their secondary schools, they reported moderate to high confidence levels in managing mental health disorders. This study suggests that SSATs develop new policies or guidelines in the areas of lowest preparedness and confidence to provide mental health care to minor student–athletes. SSATs can look to the NATA guidelines and the interassociation recommendations for mental health care to fill in the gaps at their schools. In addition, since it showed hospital-based employment structures lack the existence of policies in mental health management, SSATs in this structure are strongly encouraged to check if their institutions have mental health policies or guidelines for minors.

## Figures and Tables

**Table 1 ijerph-21-00577-t001:** Demographics.

		Frequency	Percentage (%)
Age	Number (*n*)	155	
	Mean ± SD (y/o)	40.7 ± 12.3	
	Min–Max (y/o)	23–78	
Gender (*n*, %)	Female	107/171	62.6
	Male	63	36.8
	Non-Binary	1	0.6
Ethnicity (*n*, %)	White or European American	126/171	50.8
	Hispanic, Latin (a/o), Latinx	21	8.5
	Black/African American	19	7.7
	Asian/Asian American	10	4.0
	American Indian/Alaskan	4	1.6
	Pacific Islander	4	1.6
	Prefer not to answer	1	0.4
	Prefer to self-describe	1	0.4
Year of experience	Number (*n*)	170	
	Mean ± SD (Years)	16.8 ± 11.4	
	Min–Max (Years)	1–47	

**Table 2 ijerph-21-00577-t002:** Presence and percent confidence of implementation of recommendations.

Policy, Procedures, or Guidelines Do You Have a Policy, Procedure, or Guideline Related to:	Overall Presence (Mode)	Overall Confidence (Mean ± SD)	School-Based System Presence (Yes: *n*, %)	Hospital-Based System Presence (Yes: *n*, %)	School-Based System Confidence (Mean ± SD)	Hospital-Based System Confidence (Mean ± SD)
1. Referring student–athletes to appropriately credentialed mental healthcare providers *	Yes, 40.9%	73.0 ± 25.1	*n* = 50/113, 44.2%	*n* = 20/58, 34.5%	75.2 ± 24.9	68.6 ± 25.1
2. Reviewing mental illness in student–athletes to recognize potential psychological concerns	No, 54.7%	54.1 ± 25.4	*n* = 28/112, 25%	*n* = 9/58, 15.5%	53.1 ± 26.9	56.0 ± 22.3
3. Considering unique stressors and triggers with student–athletes to recognize potential mechanisms or events that may create a mental illness or exacerbate an existing mental illness	No, 58.5%	57.1 ± 26.6	*n* = 22/113, 19.5%	*n* = 10/58, 17.2%	56.7 ± 27.3	57.7 ± 25.3
4. For stakeholders, monitoring strategies to recognize concerning behavior changes that could be related to mental health special considerations (e.g., changes in eating and sleeping habits, unexplained weight loss or weight gain) *	No, 51.5%	60.5 ± 24.7	*n* = 31/113, 27.4%	*n* = 8/58, 13.8%	61.4 ± 25.4	58.6 ± 23.3
5. For stakeholders, monitoring strategies related to student–athletes with mental health special considerations or circumstances (e.g., psychological response to injury, concussions, substance and alcohol abuse, attention-deficit hyperactivity disorder (ADHD) diagnosis, eating disorders, bullying and hazing consideration) *	No, 49.4%	58.4 ± 24.9	*n* = 36/112, 32.1%	*n* = 6/58, 10.3%	60.1 ± 26.3	55.1 ± 21.9
6. The legal implications in developing a plan to deal with the psychological concerns of student–athletes, particularly minors	No, 45.3%	46.6 ± 26.1	*n* = 20/112, 17.9%	*n* = 5/58, 8.6%	47.7 ± 27.1	44.7 ± 24.5
7. The education of stakeholders on the importance of psychological health *	No, 47.6%	59.3 ± 26.8	*n* = 31/112, 27.7%	*n* = 8/58, 13.8%	59.3 ± 27.6	59.4 ± 25.3
8. The education of stakeholders on the importance of having resources available to promote psychological health (e.g., stress-management strategies, youth mental health services, or resources within the community) *	No, 45.6%	61.2 ± 27.3	*n* = 39/113, 34.5%	*n* = 7/58, 12.1%	62.2 ± 27.4	59.3 ± 27.1
9. The collaboration between stakeholders in developing a plan to address student–athlete psychological concerns effectively	No, 47.4%	63.2 ± 26.8	*n* = 28/113, 24.8%	*n* = 12/58, 20.7%	66.4 ± 25.8	57.1 ± 27.9
10. Having a policy that includes considerations for the county, state, and federal laws and regulations regarding mandatory reporting *	Yes, 67.3%	77.5 ± 25.5	*n* = 85/113, 75.2%	*n* = 30/58, 51.7%	79.9 ± 24.9	72.6 ± 26.2
11. Mandatory reporting in cases in which an individual poses a risk to themself or others in any way *	Yes, 71.9%	80.5 ± 22.1	*n* = 89/113, 78.8%	*n* = 34/58, 58.6%	81.1 ± 23.3	79.4 ± 19.4
12. Mandatory reporting in cases in which an individual is being abused or neglected in any way *	Yes, 80.5%	81.7 ± 20.6	*n* = 98/113, 86.7%	*n* = 38/56, 67.9%	83.2 ± 22.0	78.4 ± 16.9
13. Including parental/guardian rights and notification practices regarding student–athlete mental health considerations	Yes, 34.9%	60.6 ± 28.3	*n* = 44/113, 38.9%	*n* = 15/56, 26.8%	61.2 ± 29.5	59.3 ± 26.0
14. The chain of command and contact protocol for the campus crisis intervention team regarding student–athlete mental health considerations *	Yes, 56.9%	70.3 ± 26.8	*n* = 72/111, 64.9%	*n* = 23/56, 41.1%	73.5 ± 27.1	63.7 ± 25.1
15. Regarding the training of stakeholders in recognizing a variety of potential traumatic stressors that may cause a traumatic stress reaction or violence in student–athletes *	No, 39.5%	59.0 ± 27.4	*n* = 33/106, 31.3%	*n* = 8/56, 14.3%	60.9 ± 28.3	55.4 ± 25.7
16. Regarding the referral of student–athletes who exhibit potential traumatic stressors that may cause a traumatic stress reaction or violence to credential mental healthcare providers *	Yes, 38.6%	65.3 ± 28.6	*n* = 47/103, 45.6%	*n* = 14/55, 25.5%	66.5 ± 28.7	62.8 ± 28.5
17. A recognition and referral pathway for mental health emergencies that serves as your action plan when an emergency occurs *	Yes, 48.7%	68.6 ± 27.0	*n* = 57/103, 55.3%	*n* = 20/55, 36.4%	71.8 ± 27.4	62.4 ± 25.4

*: Significant differences between employment structures were observed in the presence of recommended policies.

## Data Availability

The raw data supporting the conclusions of this article will be made available by the authors on request.

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
