# Peer review of "Exploring Secondary School Athletic Trainers’ Perspectives in Managing Mental Health Situations"

_ijerph, 2024, doi:10.3390/ijerph21050577_

Round 1

Reviewer 1 Report

Comments and Suggestions for Authors

First of all, thank you for the opportunity to evaluate this work.

I believe that the present paper complies with the general requirements for the proper elaboration of a quality research paper.

All chapters are correctly organized and structured both from the perspective of form and content.

My only suggestion to the authors is to explain and delimit the rather synthetic concept in the Introduction to what the notion of mental health refers to.

The concept is quite broad, are we referring to the respective students in secondary school, to mild anxiety, stress, or are we talking about situations in which excessive nervousness combined with aggressive behavior manifests itself, maybe even psychosis?

Author Response

Comment 1- I believe that the present paper complies with the general requirements for the proper elaboration of a quality research paper.

Comment 2- All chapters are correctly organized and structured both from the perspective of form and content.

Response: Thank you for your feedback.

Comment 3- My only suggestion to the authors is to explain and delimit the rather synthetic concept in the Introduction to what the notion of mental health refers to. The concept is quite broad, are we referring to the respective students in secondary school, to mild anxiety, stress, or are we talking about situations in which excessive nervousness combined with aggressive behavior manifests itself, maybe even psychosis?

Response: Substantial revisions have been made to the introduction. Addressed in lines 54-69

Reviewer 2 Report

Comments and Suggestions for Authors

The study discusses a topical and interesting issue of the preparedness and confidence levels of secondary school athletic trainers in managing mental health situations.

1- title

Considering the unrepresentative sample, it would be appropriate to specify in the title that this is "an exploratory study". Current perspectives do not correspond to what the article manages to demonstrate

2_ the introduction, while clearly exposing the importance of investigating the issue in order to be able to respond to a need for psychological support for adolescents who increasingly manifest mental health problems, should be expanded. 

It is suggested to:

- include more recent national statistical surveys (the data presented in the paper refer to 2010,2011);

- extend the discussion to include what has already been demonstrated in the scientific literature by also expanding the search to non-US studies.

3- Methods

the sample is small and with a high standard deviation of the age of the participants; the years of experience also show a high variability. With respect to this variable, it would have been interesting to compare the less experienced trainers with those with more years of experience. 

the confidence range to be self-reported appears excessively wide (0- 100)

the results do not provide a complete view of what was found in the questionnaire. The items presented are limited to one section of the quest. no reason is given as to why this choice was made.

4- The discussion could be better supported by other scientific sources to support the discussion of the data with a critical analysis; e.g. by providing information on other countries where mental health support services are better organised or specific guidelines exist.

it is suggested that the information requested be supplemented in order to provide a more complete framework of analysis that can highlight the reasons why policies and improvement actions need to be increased awareness.

Author Response

Comment 1- title

Considering the unrepresentative sample, it would be appropriate to specify in the title that this is "an exploratory study". Current perspectives do not correspond to what the article manages to demonstrate

Response: Revised to Exploring Secondary School Athletic Trainers’ Perspectives in Managing Mental Health Situations

Comment 2: the introduction, while clearly exposing the importance of investigating the issue in order to be able to respond to a need for psychological support for adolescents who increasingly manifest mental health problems, should be expanded. 

It is suggested to:

- include more recent national statistical surveys (the data presented in the paper refer to 2010,2011);

- extend the discussion to include what has already been demonstrated in the scientific literature by also expanding the search to non-US studies.

Response: Addressed in lines 45-50 and 66-68.

Comment 3: Methods

Comment 3-1: the sample is small and with a high standard deviation of the age of the participants; the years of experience also show a high variability. With respect to this variable, it would have been interesting to compare the less experienced trainers with those with more years of experience. 

Response: Addressed in lines 344-347.

Comment 3-2: the confidence range to be self-reported appears excessively wide (0- 100)

Response: This is consistent with confidence scales in previous studies and a citation is provided.

Comment 3-3: the results do not provide a complete view of what was found in the questionnaire. The items presented are limited to one section of the quest. no reason is given as to why this choice was made.

Response: The results narrative may be brief, but the entirety of the data set is provided in the tables. 

Comment 4: The discussion could be better supported by other scientific sources to support the discussion of the data with a critical analysis, e.g., by providing information on other countries where mental health support services are better organized or specific guidelines exist.

Response: Given the use of specific guidelines to the US, this is not the scope of this investigation.

Comment 4-1: It is suggested that the information requested be supplemented to provide a more complete framework of analysis that can highlight the reasons why policies and improvement actions need to be increased in awareness.

Response: Thank you for your feedback. We have made several of your suggested changes.

Reviewer 3 Report

Comments and Suggestions for Authors

The idea of this study is interesting. My recommendations are the following:

Abstract – it is a bit long, I recommend restricting the test. I recommend writing keywords in capital letters.

Line 94 – mention that it is a project, I recommend clarification.

Line 106 I recommend to mention in parenthesis what the 2 numerical values represent (arithmetic mean and standard deviation. I also recommend to mention a decimal to the presented values.

I recommend that table 1 be moved to the Participants section. Also to clarify why in the Age indicator, only 155 subjects are mentioned, while 171 appear in the text. You mentioned that the maximum age is 78, haven't they been retired for a few years? In this regard, I recommend to mention the inclusion criteria of the subjects, in the Participants section, and to clarify this aspect of age. Also in table 1 under the Year of experience parameter, mention 170, I recommend clarification because the total number is 171, did that person have no experience?

The introduction of bibliographic indexes is mentioned before the end of the sentence, I recommend observing the journal editing rules.

Lines 296-297 recommend deletion.

Author Response

Comment 1: Abstract – it is a bit long, I recommend restricting the test. I recommend writing keywords in capital letters.

Response: This is inconsistent with formatting requirements of the journal.

Comment 2: Line 94 – mention that it is a project, I recommend clarification.

Response: Unclear what the ask is here.

Comment 3: Line 106 I recommend to mention in parenthesis what the 2 numerical values represent (arithmetic mean and standard deviation. I also recommend to mention a decimal to the presented values.

Response: This is consistent with scientific reporting. No changes made.

Comment 4: I recommend that table 1 be moved to the Participants section.

Response: Revised as suggested.

Comment 5: Also to clarify why in the Age indicator, only 155 subjects are mentioned, while 171 appear in the text. You mentioned that the maximum age is 78, haven't they been retired for a few years? In this regard, I recommend to mention the inclusion criteria of the subjects, in the Participants section, and to clarify this aspect of age. Also in table 1 under the Year of experience parameter, mention 170, I recommend clarification because the total number is 171, did that person have no experience?

Response: Partial data were used for analysis. This has been added to the Statistical Analysis section.

Comment 6: The introduction of bibliographic indexes is mentioned before the end of the sentence, I recommend observing the journal editing rules.

Response: Thank you for the feedback; we have complied with journal reporting requirements, but in the event the sentence has clauses that require specific citations, they are included where appropriate.

Comment 7: Lines 296-297 recommend deletion.

Response: Thank you for the recommendation. We have chosen not to delete the statements as they are consistent with our findings and the recommendations of the guidelines from which we drew the purpose of the study.  

Reviewer 4 Report

Comments and Suggestions for Authors

The manuscript investigates the role of Secondary School Athletic trainers in mental health. The topic is important, but the background and the methods of the study need to be clarified. 

Comments on the Quality of English Language

The manuscript has an interesting approach, but several things need to clarify. 

·      The role of Secondary School Athletic training on mental health is more complex than it was written in the introduction. Hence, I recommend to the authors to add more studies on the role of sport, and physical activity on mental health. I also recommend briefly explaining the psychosocial development through sport. 

·      I recommend adding more information on National Athletic Trainers' Association

·      Add more procedures in participants and procedure sections

· Lines 107 and 108 should be in the statistical analysis section

·      I suggest adding examples for the instruments

·      Scaling from 0 to 100 I believe it's not a Likert-type scale. Please provide a reference for this scaling. 

·      I recommend adding Table 1 to the participants and procedure sections in the methods 

·      Provide statistical test in Table 2

Author Response

Comment 1:  The role of Secondary School Athletic training on mental health is more complex than it was written in the introduction. Hence, I recommend to the authors to add more studies on the role of sport, and physical activity on mental health. I also recommend briefly explaining the psychosocial development through sport. 

Response: Psychological component of sport addressed in lines 68-72. Role of SSATs in mental health situations expanded in lines 82-87.

Comment 2: I recommend adding more information on National Athletic Trainers' Association

Response: Addressed in line 116.

Comment 3: Add more procedures in participants and procedure sections, Lines 107 and 108 should be in the statistical analysis section

Response: Revised as suggested.

Comment 4: I suggest adding examples for the instruments

Response: The instrument is provided in table 2.

Comment 5: Scaling from 0 to 100 I believe it's not a Likert-type scale. Please provide a reference for this scaling. 

Response: References have been added.

Comment 6: I recommend adding Table 1 to the participants and procedure sections in the methods 

Response: Revised as suggested.

Comment 7: Provide statistical test in Table 2

Response: This is detailed in the Statistical Analysis Section.